# Human Wnt/β-Catenin Regulates Alloimmune Signaling during Allogeneic Transplantation

**DOI:** 10.3390/cancers13153798

**Published:** 2021-07-28

**Authors:** Mahinbanu Mammadli, Rebecca Harris, Sara Mahmudlu, Anjali Verma, Adriana May, Rohan Dhawan, Adam T. Waickman, Jyoti Misra Sen, Avery August, Mobin Karimi

**Affiliations:** 1Department of Microbiology and Immunology, SUNY Upstate Medical University, Syracuse, NY 13210, USA; MammadlM@upstate.edu (M.M.); WhitmanR@upstate.edu (R.H.); mahmudls@uptate.edu (S.M.); MayA@upstate.edu (A.M.); rdhawan@u.rochester.edu (R.D.); WaickmaA@upstate.edu (A.T.W.); 2Biomedical Research Center, National Institute on Aging-National Institutes of Health, 08C218, 251 Bayview Boulevard, Suite 100, Baltimore, MD 21224, USA; anjali.verma2@nih.gov (A.V.); senjy@grc.nia.nih.gov (J.M.S.); 3Immunology Program, Department of Medicine, Johns Hopkins School of Medicine, Baltimore, MD 21224, USA; 4Department of Microbiology and Immunology, College of Veterinary Medicine, Cornell University, Ithaca, NY 14853, USA; averyaugust@cornell.edu

**Keywords:** Wnt/β-catenin, GVHD, GVT, Cytokines, Inflammation, Migration, T cells signaling

## Abstract

**Simple Summary:**

This manuscript showed that Wnt/β-catenin plays a significant role in T cell-mediated GVHD after allogeneic transplantation. Our functional and genetic data demonstrated that the Wnt/β-catenin pathways play a central role in uncoupling GVHD from GVL functions.

**Abstract:**

Allogeneic hematopoietic stem cell transplantation (allo-HSCT) is one of the most widely applied forms of adoptive immunotherapy for the treatment of hematological malignancies. Detrimental graft-versus-host disease (GVHD), but also beneficial graft-versus-leukemia (GVL) effects occurring after allo-HSCT are largely mediated by alloantigen-reactive donor T cells in the graft. Separating GVHD from GVL effects is a formidable challenge, and a greater understanding of donor T cell biology is required to accomplish the uncoupling of GVHD from GVL. Here, we evaluated the role of β-catenin in this process. Using a unique mouse model of transgenic overexpression of human β-catenin (*Cat-Tg*) in an allo-HSCT model, we show here that T cells from *Cat-Tg* mice did not cause GVHD, and surprisingly, *Cat-Tg* T cells maintained the GVL effect. Donor T cells from *Cat-Tg* mice exhibited significantly lower inflammatory cytokine production and reduced donor T cell proliferation, while upregulating cytotoxic mediators that resulted in enhanced cytotoxicity. RNA sequencing revealed changes in the expression of 1169 genes for CD4, and 1006 genes for CD8^+^ T cells involved in essential aspects of immune response and GVHD pathophysiology. Altogether, our data suggest that β-catenin is a druggable target for developing therapeutic strategies to reduce GVHD while preserving the beneficial GVL effects following allo-HSCT treatment.

## 1. Introduction

Allogeneic hematopoietic stem cell transplantation (allo-HSCT) is a curative treatment for patients with hematological malignancies, due to the eradication of host malignant cells by donor T cells (graft-versus-leukemia or GVL) [1]. In 40–70% of patients, the same donor T cells also attack healthy tissues like the gastrointestinal (GI) tract and liver, leading to graft-versus-host disease (GVHD) [2]. The mortality rate due to GVHD is higher than 20% [3,4]. Therefore, therapeutic protocols that enhance the control of GVL and diminish the effects due to GVHD are essential for the treatment of leukemia.

Wnt/β-catenin signaling plays a critical role in T cell development and tissue homeostasis [5]. Wnt/β-catenin pathways also play important roles in thymocyte development, differentiation, polarization, and survival of mature T lymphocytes [6,7]. The transcription factor T Cell Factor-1 (*TCF-1*), encoded by the gene, and Lymphoid Enhancer Binding Factor-1 (*LEF-1*) is the downstream transcription effectors of the canonical Wnt signaling pathway [8]. Both *Tcf7* and *Lef1* are highly expressed by naïve CD8^+^ T cells, and as these cells encounter antigen, the levels of *Tcf7* and *Lef1* decrease in these cells [9]. There are several lines of evidence suggesting that Wnt/β-catenin signaling downregulates the production of proinflammatory cytokines [10]. These cytokines include IL-1β, IL-6, IL-8, and TNF-α. Wnt/β-catenin signaling is also critical for T cell differentiation, effector functions, and migration [11]. The activation of β-catenin signaling turns CD8^+^ T cells into Tfh-like cells [12]. β-catenin signaling can also enhance the differentiation of CD8^+^ T cells into effector cells [13]. Wnt/β-catenin signaling pathway components are highly expressed in naive CD8^+^ and memory CD8^+^ T cells, but expressed less in effector CD8^+^ T cells [11]. Since the activation and maintenance of T cells are both required for anti-tumor immunity (GVL) and GVHD, we hypothesized that Wnt/β-catenin signaling may play an important role in these linked processes. 

In this study, we used a mouse model expressing transgenic β-catenin (*Cat-Tg* mice), that increases the expression of the protein by 2–3 fold [8]. We demonstrate that donor T cells from *Cat-Tg* mice do not induce GVHD in an MHC-mismatched mouse model, but still clear primary B-cell acute lymphoblastic leukemia (B-ALL) cells [14,15]. Our data also showed that *Cat-Tg* mice had fewer naïve CD8^+^ T cells and an increase in T cells with an activated phenotype that may trend towards exhaustion. Interestingly, our data show that recipient mice allogeneically transplanted with T cells from *Cat-Tg* mice had significantly decreased proinflammatory cytokines in serum, and the donor T cells showed lower levels of expansion. Unbiased analysis of gene expression using RNA sequencing showed that transgenic expression of β-catenin affects pathways like regulation of metabolism, lymphocyte activation, adaptive immune response, MHC complex, cytokine production, antigen processing and presentation, Th1/Th2 cell differentiation, Th17 cell differentiation, cell adhesion molecules, chemokine signaling, signaling (NF-κβ, TNF and others), GVHD, allograft rejection, and autoimmunity in CD8^+^ and CD4^+^ T cells. Importantly, trafficking of donor T cells to GVHD target organs is considered a hallmark of GVHD [16]. Our data showed that transgenic expression of β-catenin specifically affects CD8^+^ T cell migration to the GVHD target organs after allo-BMT. We examined tissue damage to target organs using histology, and recipient mice transplanted with donor T cells from *Cat-Tg* mice showed significantly less damage compared to recipient mice transplanted with donor T cells from WT mice. In summary, we provide a mechanistic understanding of the manner in which enhancement of the Wnt/β-catenin-TCF1/LEF1 pathway protects from GVHD while maintaining GVL. For the first time, we provide evidence that β-catenin has a significant impact on T cell functions in an allotransplant model. This effect is due to changes in T cell phenotype, function, and gene expression. Thus, we show that β-catenin and Wnt signaling are potential druggable targets to separate GVHD and GVT to improve allo-HSCT outcomes.

## 2. Results

### 2.1. Donor T Cells from Cat-Tg Mice Do Not Induce GVHD but Maintain GVL Function

To determine whether overexpression of β-catenin impacts GVHD pathogenesis after allo-HSCT, we examined the effects of β-catenin signaling on donor CD4^+^ and CD8^+^ T cells in an allotransplant model, using C57Bl/6 background mice (MHC haplotype^b^) as donors and BALB/c mice (MHC haplotype ^d^) as recipients [15]. To induce GVHD, we used MHC-mismatched donors and recipients. T cell-depleted bone marrow cells from WT mice, and T cells from C57BL/6 (B6) WT or *Cat-Tg* mice were injected into irradiated BALB/c mice along with luciferase-expressing B-cell acute lymphoblastic leukemia (B-ALL-*luc)* tumor cells [14,15]. Lethally irradiated BALB/c mice were injected intravenously with 1 × 10^7^ wild-type (WT) T cell-depleted donor BM cells, 1 × 10^6^ MACS-sorted donor CD3^+^ T cells, and 1 × 10^5^ B-ALL-*luc* blast cells as described [14,15,17]. Recipient BALB/c mice were monitored for cancer cell growth using IVIS bioluminescence imaging for over 60 days (Figure 1A) [15,17]. While leukemia cell growth was observed in mice given bone marrow without T cells, leukemia cell growth was not seen in mice transplanted with T cells from either WT or *Cat-Tg* mice. As expected, mice transplanted with WT T cells cleared the leukemia cells (Figure 1A) but suffered significantly from GVHD (Figure 1B–D). In contrast, mice transplanted with *Cat-Tg* T cells cleared the leukemia cells (Figure 1A) and displayed minimal signs of GVHD (Figure 1B–D). All animals transplanted with *Cat-Tg* T cells survived for more than 65 days post-allo-HSCT (Figure 1B), with significantly reduced weight loss and clinical scores compared to those transplanted with WT T cells (scored based on weight loss, posture, activity, fur texture, and skin integrity, and diarrhea as previously described) [18] (Figure 1C,D). Quantification of tumor bioluminescence showed that mice given WT or *Cat-Tg* T cells cleared the tumor cells, while tumor burden remained high for mice only given bone marrow (Figure 1E). Our results indicate that donor T cells from *Cat-Tg* mice are able to generate anti-leukemia immunity, without inducing GVHD. 

### 2.2. Wnt/β-Catenin Affects T Cell Phenotype and Cytotoxic Function

To examine how the overexpression of Wnt/β-catenin affects T cell phenotype, we MACS-purified T cells from *Cat-Tg* and WT mice by CD90.2-positive selection. We examined the effects of Wnt/β-catenin overexpression on CD4^+^ and CD8^+^ T cells in comparison to T cells from WT C57Bl/6 mice. Our data showed that CD8^+^ T cells from *Cat-Tg* mice exhibit an innate memory phenotype (IMP) [15,19], as indicated by the expression of high levels of CD44, CD122, and a key transcription factor Eomesodermin (Eomes). We did not observe any changes in expression of T-box transcription factor TBX21, also called T-bet. CD8^+^ T cells from *Cat-Tg* mice showed a trend towards increased levels of PD-1, but had no differences in CTLA-4 expression compared to CD8^+^ T cells from WT mice. We also did not observe significant differences in TCF-1 expression (Figure 2A,B). Analysis of the effect of Wnt/β-catenin on CD4^+^ T cells indicated that CD4^+^ T cells from *Cat-Tg* mice also express higher levels of CD44 but had no differences in CD122, Eomes, or T-bet expression compared to CD4^+^ T cells from WT mice. CD4^+^ T cells from *Cat-Tg* mice also express significantly higher percentages of PD-1 but have no differences in CTLA-4 or TCF-1 compared to CD4^+^ T cells from WT mice (Figure 2C,D). Our data show that CD4^+^ T cells from *Cat-Tg* mice had no difference in memory phenotypes compared to CD4^+^ from WT mice (Figure 2E) (Appendix A). Next, we examined whether CD8^+^ T cells from *Cat-Tg* mice exhibited changes in memory subsets. We observed a significant decrease in naïve CD8^+^ T cells, and no significant differences in the transitioning/activating cells, but significantly increased central memory CD8^+^ T cells from *Cat-Tg* mice. No significant differences were observed in effector memory CD8^+^ T cells between *Cat-Tg* and WT mice (Figure 2F) (Appendix A). 

To examine whether T cells from *Cat-Tg mice* expressed proteins involved in cytotoxic function, we purified CD3^+^ T cells from *Cat-Tg* mice and WT mice, and performed a Western blot on the cell lysates. Our Western data show that CD3^+^ T cells from *Cat-Tg* mice express significantly higher levels of granzyme B and perforin than CD3^+^ T cells from WT mice (Figure 2G–I, quantified in Figure 2H,I). Next, we examined whether CD8^+^ T cells from *Cat-Tg* mice could mount a cytotoxic response, using a cytotoxicity assay against primary B-ALL cells with different effector to target ratios [14,15,17]. We found that CD8^+^ T cells from *Cat-Tg* mice effectively killed significantly more primary leukemia cells in vitro than CD8^+^ T cells from WT mice (Figure 2J). Our findings demonstrate that CD8^+^ T cells from *Cat-Tg* mice have enhanced activation markers, significantly altered CD8^+^ T cells phenotype, enhanced expression of Granzyme B and Perforin, and exert better cytotoxicity against primary leukemia cells than CD8^+^ T cells from WT mice. 

### 2.3. Wnt/β-Catenin Overexpression Results in Reduced Cytokine Production and Donor T Cell Proliferation without Affecting TCR Signaling

The conditioning regimen for allo-HSCT elicits an increase in the production of inflammatory cytokines by donor T cells, known as a “cytokine storm” [20,21]. This is considered one of the hallmarks of GVHD pathogenesis [22]. We assessed cytokine production by *Cat-Tg* T cells in our allo-HSCT model (B6→BALB/c) by examining the levels of serum inflammatory cytokines. We observed that recipient BALB/c mice treated with 1 × 10^6^ CD3^+^ T cells from *Cat-Tg* mice expressed significantly less IFN-γ, TNF-α, IL-5, IL-2, IL-6, IL-10 and IL-22 in serum compared to recipient BALB/c mice treated with CD3^+^ T cells from WT mice (Figure 3A). We did not observe differences in IL-4, IL-9, IL-17A, IL-17F, and IL-13 on day 7 post allotransplantation (Figure 3A). We also examined donor CD8^+^ or CD4^+^ T cells from secondary lymphoid organs of recipients using anti-H2K^b^ antibodies (H2K^b^ is expressed by donor C57Bl/6 cells). Ex vivo donor T cells were stimulated with anti-CD3/CD28 antibodies for 6 h in the presence of GolgiPlug (Figure 3B,C) or left unstimulated, followed by an analysis of IFN-γ and TNF-α cytokine production. *Cat-Tg* CD8^+^ or CD4^+^ T cells produced significantly less inflammatory IFN-γ when stimulated via anti-CD3/CD28 antibodies, but we did not observe any differences in TNF-α expression (Figure 3B–E). 

We next examined donor T cell proliferation using an EdU incorporation assay. We utilized short-term allo-transplantation as described above, and recipients were injected with EdU in PBS on days 5 and 6 post-transplant. Seven days post allo-transplantation, splenocytes were obtained from recipients, and donor cells (identified by H2K^b+^, CD3^+^ and CD4^+^ or CD8^+^) were examined for proliferation by EdU incorporation. Both donor CD4^+^ and CD8^+^ T cells from *Cat-Tg* mice showed a trend toward reduced proliferation compared to donor T cells from WT mice, but this effect was not significant (Figure 3F–H). 

Attenuated TCR signaling due to the absence of ITK causes T cells to acquire an innate-like memory phenotype (IMP), distinguished by higher expression of CD44, CD122 and Eomes [15,17,19]. The absence of ITK also affects T cell receptor-induced phospho-ERK, phospho-PLCγ-1 and expression levels of IRF-4 [15]. However, despite the similarity of the observed T cell phenotype to the absence of ITK, we did not observe significant differences in any of these signaling molecules on T cells from *Cat-Tg* or WT mice on the total protein level (Figure 3I,J). Our data suggest that donor T cells from *Cat-Tg* mice exhibit reduced inflammatory cytokine production and reduced proliferation upon allogeneic transplantation in a major mismatch model. These findings support our observations that donor T cell-induced GVHD severity is reduced by overexpression of β-catenin.

### 2.4. Wnt/β-Catenin Expression Regulates Gene Expression in T Cells during GVHD

As an unbiased approach to further explore differences between CD4^+^ or CD8^+^ T cells from WT and *Cat-Tg* mice, we employed RNA sequencing analysis. We examined the differences in gene expression between WT and *Cat-Tg* CD4^+^ and CD8^+^ T cells before and following allo-HSCT. We sort-purified donor WT and *Cat-Tg* CD4^+^ or CD8^+^ T cells from freshly isolated splenocytes, and called these pre-transplanted cells (pre-tx). We also MACS purified 1 × 10^6^ CD4^+^ and 1 × 10^6^ CD8^+^T cells, which were mixed at a 1:1 ratio, from WT or *Cat-Tg* mice and transplanted them along with T cell-depleted bone marrow cells into the tail vein of lethally irradiated BALB/c recipients. On day 7 post-transplantation, we sort-purified donor WT and *Cat-Tg* CD4^+^ or CD8^+^ T cells (using H-2K^b^ antigen expressed by donor T cells) from recipients, and called these post-transplant day 7 samples (post-tx). The cells were sorted into Trizol reagent and transcriptionally profiled. Principal component analysis (PCA) of CD8^+^ T cells identified four clusters of samples, which clearly separated the pre-transplanted WT or *Cat-Tg* and post-transplanted WT or *Cat-Tg* populations along PC1 (32%), PC2 (15%) and PC3 (13%) (Figure 4A). Further analysis of pre-transplanted CD8^+^ T cell populations identified 1006 differentially expressed genes (DEGs; FDR ≤ 0.05, log Fold Change logFC ≥ 2) between WT and *Cat-Tg* (Figure 4B), of which 624 genes were downregulated and 382 genes were upregulated (Figure 4C). The use of a Spearman correlation method associated with hierarchical clustering categorized pre tx CD8^+^ T cell samples into two clusters, WT and *Cat-Tg* (Figure 4D). DEGs between WT and *Cat-Tg* in pre-transplanted CD8^+^ T cells were averaged by group and gene co-regulation was determined by hierarchical clustering, using Pearson correlation with a grouping cutoff (*k*) of 2. Genes that were up and downregulated in Cat-Tg pre-tx CD8^+^ T cell samples clustered in Module 1 and Module 2, respectively. Gene expression is averaged by group (*n* = 3) for clarity and displayed as *z* score across each row (Figure 4D). Go enrichment analysis of Module 1 revealed that upregulated DEGs are involved in numerous biological pathways including peptide metabolism, metabolic process, PD1 signaling, and downstream TCR signaling. Go enrichment analysis of downregulated DEGs in Module 2 revealed that these genes are involved in numerous biological pathways including antigen processing and presentation, response to cytokine, cytokine receptor activity, protein phosphorylation, protein tyrosine kinase activity, NF-κappa B signaling, Th17 differentiation, Interleukin 17 signaling, Th1 and Th2 differentiation, cell adhesion molecules, chemokine signaling, TLR signaling, MAPK signaling, apoptosis, innate immune system, and a number of diseases like Allograft rejection, Graft-versus-host disease, Inflammatory bowel disease, Type I Diabetes, Systemic Lupus Erythematosus, and Autoimmunity (Figure 4E). When we performed GSEA analysis using the Hallmark pathways collection from Molecular Signatures Database (MSigDB) [23] we observed that Hallmark of TNF-α signaling via NF-κβ, KRAS signaling, Interferon-gamma response, Inflammatory response, IL-6 JAK STAT3 Signaling, IL-2 STAT2 signaling, G2M checkpoint, Apical junction, and allograft rejection pathways were enriched in WT compared to *Cat-Tg* in pre-transplanted CD8^+^ samples (Figure 4F–I). Pathways including DNA repair and oxidative phosphorylation were enriched in to *Cat-Tg* compared to WT (Figure 4J). 

The analysis of the post-transplanted CD8^+^ T cell samples cell populations identified 227 differentially expressed genes (DEGs; FDR ≤ 0.05, logFC ≥ 2) between WT and *Cat-Tg* (Figure 5A), of which 46 genes were downregulated and 181 genes were upregulated (Figure 5B). Post-transplanted CD8^+^ T cell samples were categorized into two clusters, WT and *Cat-Tg* (Figure 5C). DEGs between WT and *Cat-Tg* in post-transplanted CD8^+^ T cells again were averaged by group and gene co-regulation was determined by hierarchical clustering. Genes that were up- and downregulated in Cat-Tg post-tx CD8^+^ T cell samples clustered in Module 1 and Module 2, respectively (Figure 5C). Go enrichment analysis of Module 2 revealed that downregulated DEGs are involved in viral protein interaction with cytokine and cytokine receptor, chemokine signaling, chemokine receptors bind chemokines and others (Figure 5D). GSEA analysis showed that Hallmark of Interferon gamma and alpha response, Inflammatory response, and Apoptosis pathways were enriched in *Cat-Tg* compared to WT in post-transplanted CD8^+^ samples (Figure 5E,F). Pathways including Coagulation and KRAS Signaling DN were enriched in WT compared to *Cat-Tg* in post-transplanted CD8^+^ samples (Figure 5E,G). 

Principal component analysis (PCA) analysis of the CD4^+^ T cell samples identified four clusters of samples, which again clearly separated the pre-transplanted WT or *Cat-Tg* and post-transplanted WT or *Cat-Tg* populations along PC1 (39.5%), PC2 (17%) and PC3 (10.3%) (Figure 6A). Further analysis of pre-transplanted CD4^+^ T cell populations identified 1169 differentially expressed genes (DEGs; FDR ≤ 0.05, log Fold Change logFC ≥ 2) between WT and *Cat-Tg* (Figure 6B), of which 387 genes were downregulated and 782 genes were upregulated (Figure 6C). The use of a Spearman correlation method associated with hierarchical clustering categorized pre-transplanted CD4^+^ T cell samples into two clusters, WT and *Cat-Tg* (Figure 6D). DEGs between WT and *Cat-Tg* in pre-transplanted CD4^+^ T cells were averaged by group and gene co-regulation was determined by hierarchical clustering. Genes that were up and downregulated in Cat-Tg pre-tx CD4^+^ T cell samples clustered in Module 1 and Module 2 respectively (Figure 6D). Go enrichment analysis of Module 1 revealed that upregulated DEGs are involved in numerous biological pathways including peptide metabolic process, metabolism, PD1 signaling, TNF receptor binding, oxidative phosphorylation, TCA cycle, Respiratory electron transport, and genetic transcription pathway. Go enrichment analysis of Module 2 revealed that downregulated DEGs are involved in pathways including immune receptor, cytokine receptor activity, antigen processing and presentation, lymphocyte activation, cytokine mediated signaling, MHCII protein complex, cell adhesion molecules, Th17 differentiation, NF-kappa B signaling, Th1 and Th2 differentiation, innate immune system, hemostasis, IL-5 Signaling, IRF-4–Spi1 complex and number of diseases like Primary immunodeficiency, Asthma, Rheumatoid arthritis, Allograft rejection, Graft-versus-host disease, Inflammatory bowel disease, Type I Diabetes, and Autoimmunity (Figure 6E). GSEA analysis using the Hallmark pathways collection from Molecular Signatures Database (MSigDB) [23] showed that Hallmark of TNF-α signaling via NF-kβ, Apical junction, Coagulation and Hypoxia were enriched in WT compared to *Cat-Tg* in pre-transplanted CD4^+^ samples (Figure 6F,G). Finally, pathways including MYC targets and oxidative phosphorylation were enriched in *Cat-Tg* in pre-transplanted CD8^+^ samples (Figure 6F,H). 

The analysis of the post-transplanted CD4^+^ T cell samples identified 761 differentially expressed genes (DEGs; FDR ≤ 0.05, logFC ≥ 2) in comparison between *Cat-Tg* and WT (Figure 7A), of which 314 genes were downregulated and 447 genes were upregulated **(**Figure 7B). Post-transplanted CD4^+^ T cell samples were categorized into two clusters, WT and *Cat-Tg* (Figure 7C). Genes that were downregulated and upregulated in Cat-Tg post-tx CD4^+^ T cell samples clustered in Module 1 and Module 2, respectively (Figure 7C). GO enrichment analysis of Module 1 revealed that dowregulated DEGs are involved in viral protein interaction with diacylglycerol pathway, hemostasis, metabolic process, post-translational protein phosphorylation, chemokine receptors bind chemokines and others (Figure 7D). GO enrichment analysis of Module 2 revealed that upregulated DEGs are involved in viral protein interaction immune response, antigen processing and presentation, cell adhesion molecules, cytokine–cytokine receptor interaction, hemostasis and some disorders like Graft-versus-host disease, Inflammatory bowel disease, Allograft rejection and others (Figure 7D). GSEA analysis showed that Hallmark of KRAS signaling DN, G2M checkpoint, E2F targets. Coagulation pathways were enriched in WT compared to *Cat-Tg* in post-transplanted CD4^+^ samples (Figure 7E,F). Further, pathways including Apoptosis, Il-6 JAK STAT3 signaling, Inflammatory response, Interferon alpha and gamma response, and TNF-a signaling via NF-kappa b were enriched into *Cat-Tg* compared to WT in post-transplanted CD4^+^ samples (Figure 7G). Altogether these data suggest that the genetic signature of CD8^+^ and CD4^+^ T cells that overexpress β-catenin is different from CD8^+^ and CD4^+^ T cells from WT mice. 

### 2.5. Wnt/β-Catenin Overexpression Specifically Affects CD8^+^ T Cell Functions

The pathogenesis of GVHD involves the migration of donor T cells into the target organs in the recipient, including the liver, small intestine, and skin [24,25]. GVHD occurs in a subset of organs and involves early migration of alloreactive T cells into these organs followed by T cell expansion and later tissue destruction [16,26]. To examine whether overexpression of Wnt/β-catenin affects donor T cell migration, irradiated BALB/c recipient mice were injected with CD8^+^ T cells and CD4^+^ T cells from *Cat-Tg* (CD45.2^+^) and WT B6-Ly5.1 (CD45.1^+^) mice, mixed at a 1:1 ratio of WT: *Cat-Tg* (Figure 8A). A total of 2 × 10^6^ T cells were injected, and the cells were checked prior to transplant for a 1:1 ratio of CD4/CD8 for each strain, and for a 1:1 ratio of donor strains, CD45.1^+^ WT and CD45.2^+^
*Cat-Tg* (Figure 8B). As a control, we also transplanted WT CD45.2 (C57BL/6) and WT CD45.1 (B6-Ly5.1) cells at a 1:1 ratio of WT(CD45.2): WT(CD45.1). (Figure 8A,C). Once again, 1 × 10^6^ T cells were injected and cells were checked prior to transplant for a 1:1 ratio of CD4/CD8 for each strain, and for a 1:1 ratio of donor strains (Figure 8D). At 7 days post-transplantation, recipient mice were examined for the presence of donor T cells in the spleen and liver. Our data show that recipient mice transplanted with WT CD45.1 and WT CD45.2 cells had no difference in post-transplanted T cell ratios from spleen and liver (Figure 8E,F). When we examined the recipient, mice transplanted with WT CD45.1 and WT CD45.2 cells, they had significantly reduced donor CD4^+^ cells versus donor CD8^+^ T cells in the spleen and liver (Figure 8G,H). We did not observe any difference in CD4 and CD8 T cells from the spleen in a comparison between cells from CD45.1 versus CD45.2 mice (Figure 8G). We observed a slight decrease in CD8 T cells and a slight increase in CD4 T cells from the liver in a comparison between cells from CD45.1 versus CD45.2 mice (Figure 8H).

When we examined spleens from recipient mice transplanted with CD45.1 WT and CD45.2 *Cat-Tg* cells, we observed that the percentage of CD45.2 *Cat-Tg* T cells in the spleen and liver was significantly reduced compared to CD45.1 WT T cells (Figure 8I,J). The recipient mice transplanted with WT CD45.1 and Cat-Tg CD45.2 cells had a reduction in donor CD4^+^ T cells versus donor CD8^+^ T cells in the spleen and liver (Figure 8K,L). We did observe a significant decrease in CD8 T cells from the spleen and liver in a comparison between cells from CD45.1 versus CD45.2 mice (Figure 8K,L). When we compared CD4 T cells, we saw a significant increase in spleen and liver in a comparison between cells from CD45.1 versus CD45.2 mice (Figure 8K,L). 

Using histological staining for H&E, we also observed leukocyte infiltration into GVHD target organs like the liver and small intestine (SI) [24] in WT T cell recipients, but not as much in *Cat-Tg* T cell recipients (Figure 8M and Appendix A). These data suggest that CD8^+^ T cells from *Cat-Tg* mice have significantly been affected by β-catenin overexpression, and caused less tissue damage to GVHD target tissues. 

## 3. Discussion

In this report, we demonstrate that overexpression of Wnt/β-catenin regulates allo-reactive T cells for the treatment of hematological malignancies. CD8^+^ and CD4^+^ T cells from *Cat-Tg* mice expressed higher levels of CD44 and PD-1 markers. CD8^+^ T cells from *Cat-Tg* mice also expressed higher Eomes and CD122. We did not observe any differences in TCF-1, T-bet or CTLA-4 expression. CD8^+^ T cells from *Cat-Tg* mice showed frequencies of central memory and transitioning cells, but a reduced naïve cell population. Furthermore, both CD4^+^ and CD8^+^ T cells from mice overexpressing β-catenin (*Cat-Tg* mice) showed significantly reduced GVHD pathogenesis, while maintaining GVL in models of allo-HSCT. 

Several lines of evidence suggested that CD44^hi^ and CD122^hi^ T cells do not induce GVHD [27,28,29]. Our data showed that a high proportion of CD8^+^ T cells from *Cat-Tg* mice are CD44^hi^ and CD122^hi^ and express higher levels of Eomes (IMP phenotype) [19]. Previously, it has been suggested that the IMP phenotype might be due to higher expression of IL-4 in the thymus of *Cat-Tg* mice, which can result in the IMP phenotype [30]. However, published data have indicated that the IMP phenotype is not dependent on IL-4 expression specifically in *Cat-Tg* mice [8,31] These findings would indicate that higher Eomes expression and the IMP phenotype in *Cat-Tg* mice due to β-catenin overexpression allows these cells to have anti-tumor activity in a T cell-intrinsic manner [15,17]. 

Several lines of evidence also suggest that β-catenin plays a central role in T cell development [12,32,33]. Experiments using either loss of β-catenin or enforced expression of stabilized β-catenin have further identified a role for β-catenin at multiple stages of T cell development [33,34]. Adoptive transfer of Wnt-treated CD8^+^ T cells was shown to enhance anti-tumor activity in vivo [35]. Our data provide evidence that CD8^+^ T cells from *Cat-Tg* mice express higher levels of granzyme B and perforin, higher expression of Eomes, and these cells also exhibited enhanced cytotoxicity. Constitutive activation of the TCF-1/β-catenin pathway in vivo has been shown to favor the generation of memory CD8^+^ T cells [36]. 

To examine how T cells from *Cat-Tg* mice maintain GVL function without GVHD damage, we examined proinflammatory cytokine expression. Our data show that donor T cells from *Cat-Tg* mice express significantly less proinflammatory cytokines both on a serum level and on a cellular level in our allo-HSCT model. Transcriptome analysis by RNA sequencing revealed that there were 150 differentially expressed genes in CD4^+^ T cells and over 250 genes affected by overexpression of β-catenin in CD8^+^ T cells. Pathway analysis revealed that the differentially expressed genes in CD4^+^ T cells are involved in the regulation of immune system processes, T cell and B cell activation, T cell proliferation, adaptive immune responses, immune system development, inflammatory responses, cytokine production, signaling, cell adhesion, and chemokine receptors. Genes that were differentially expressed in CD8^+^ T cells were involved in similar pathways, along with hematopoietic cell lineage, GVHD, allograft rejection, Th1, Th2, Th17 differentiation, and other pathways. Therefore, these findings suggest that β-catenin may play a critical role in regulating gene expression programs of mature T cells during alloactivation. Taken together, these data suggest that Wnt/β-catenin could represent a potential target for the separation of GVHD and GVL responses after allo-HSCT. 

Donor T cell proliferation and migration to GVHD target organs are considered hallmarks of GVHD [4]. The Wnt/β-catenin signaling path is a highly conserved pathway through evolution, and regulates key cellular functions including proliferation, differentiation, migration, genetic stability, apoptosis, and stem cell renewal [34,36]. CD8^+^ T cells, but not CD4^+^ T cells from *Cat-Tg* mice are likely to be defective in migration to GVHD target organs. Recipient BALB/c mice were transplanted with a 1:1 ratio of WT: *Cat-Tg* T cells, and at day 7 post-transplantation, we observed that the T cells were skewed toward CD4 T cells. However, when we further examined WT:WT T cells, we observed a trend towards CD8 T cells. We observed a significant reduction in CD8^+^ T cells from *Cat-Tg* mice in the spleen and liver. Both donor CD4^+^ and CD8^+^ T cells from *Cat-Tg* mice showed a trend toward reduced proliferation compared to donor CD4^+^ and CD8^+^ T cells from WT mice. Our functional data were confirmed by RNA sequencing. GO enrichment analysis revealed that the genes involved in lymphocyte cell adhesion (like CADM1, CD22, CD40, CD40LG, H2-AA, H2-AB1, H2-DMB1, H2-DMB2, H2-EB1, H2-EB2, ICOSL, ITGAM, SDC1, SDC3, SDC4, VCAM1) and chemokine signaling (like ADRBK2, CCL6, CCR6, CX3CR1, CXCL16, CXCR5, FGR, HCK, LYN, NFKB1, PIK3R6, PRKCD, STAT2, VAV2) were downregulated in T cells from *Cat-Tg* mice. These results provide evidence that β-catenin expression might suppress donor T cell migration. Published data have shown that irradiation causes the upregulation of cell adhesion molecules and provides early costimulatory signals to incoming donor T cells in the intestine, followed by a cascade of proinflammatory signals in other organs once the alloresponse is established [37]. Our data provide evidence that both CD4 and CD8^+^ T cells from *Cat-Tg* mice show significant reductions in adhesion molecules, but CD8^+^ T cells from *Cat-Tg* mice were more significantly affected, which could affect their ability to migrate to target organs.

We also observed that genes involved in chemokine receptor activity, and several genes involved in the regulation of lymphocyte migration were affected in CD4^+^ and CD8^+^ T cells. Inflammatory chemokines are expressed in inflamed tissues by both hematopoietic and non-hematopoietic cells upon stimulation by pro-inflammatory cytokines, including TNFα and IFN-γ [38]. Inflammatory chemokines of the CC, C, or CXC3C families are also increasingly expressed after allogeneic transplantation [38]. Cellular sources of chemokines may differ between specific target organs, and contribute considerably to the severity of GVHD. Our data show that inflammatory chemokines are significantly affected in T cells from *Cat-Tg* mice, which may contribute to less severe development of GVHD. 

Recently, several lines of evidence have suggested that modulating intracellular signaling pathways that regulate T cell responses and survival can be used to inhibit T cell alloresponses, T cell survival, and thus, GVHD [39]. This has included targeting the transcription factor nuclear factor kappa B (NFκβ). Our data show that T cells (both CD4 and CD8^+^) from *Cat-Tg* mice have significant changes to the NFκβ pathways. NFκβ controls the expression of a number of genes important for mediating immune and inflammatory responses. As NFκβ has long been known to play a critical role in T cell biology, particularly with respect to cytokine responses, it has always been an attractive target [40]. Several lines of evidence have recently suggested that inhibiting NFκβ signaling ameliorates GVHD in both mice and humans [41]. Bortezomib and PS-1145 are small molecule inhibitors that have been used to treat acute GVHD [42]. 

The use of agonists to activate Wnt/β-catenin signaling could have considerable clinical implications for the improvement of immunotherapies based on ex vivo manipulation of T lymphocytes for adoptive transplantation. Several mouse models have shown that blocking Gsk-3β using small molecule inhibitors resulted in the generation of stem-like memory CD8^+^ T cells, which have the potential to be highly effective in immunotherapy [43]. Using pharmacological approaches, human CD8^+^ T cells with stem-like properties can be generated using antagonists of Wnt signaling, and can be genetically engineered to have tumor-specific properties [44]. For the first time, we showed that Wnt/β-catenin plays a significant role in T cell-mediated GVHD after allogeneic transplantation. Our functional and genetic data demonstrated that the Wnt/β-catenin pathways play a central role in uncoupling GVHD from GVL functions. 

## 4. Materials and Methods

### 4.1. Mice

*Cat-Tg* mice were described previously [45]. Briefly, the ΔN87βCat fragment has the N-terminal deletion mutant of human β-catenin gene as described [46]. The ΔN87βCat fragment was cloned with the *Bam*HI site in p1017 as described [47]. The ΔN87βCat fragment with *Not*I-cut DNA was injected in FVB recipient mice. Transgenic mice were identified by Southern blot analysis of DNA as described [48]. The probe consisted of the ΔN87βCat gene. Transgenic founders (ΔCat-1, ΔCat-2, ΔCat-3 and ΔCat-7) were bred to C57BL/6 mice and maintained as heterozygous for the transgene. Mice were bred back with back cross with C57BL/6 to obtain *Cat-Tg* mice. C57BL/6, C57BL/6.SJL (B6-SJL), B6-Ly5.1 (B6.SJL-Ptprc^a^ Pepc^b^/BoyCrl) and BALB/c mice were purchased from Charles River or Jackson Laboratory. Mice aged 8–12 weeks were used, and all experiments were performed with age and sex-matched mice. Animal maintenance and experimentation were performed in accordance with the rules and guidance set by the institutional animal care and use committees at SUNY Upstate Medical University.

### 4.2. Reagents, Cell Lines, Flow Cytometry

Monoclonal antibodies were purchased from Biolegend (San Diego, CA, USA) or eBioscience (San Diego, CA, USA) and were used at 1:100 dilution. Antibodies used included mouse anti-CD3(cat#100102), anti-CD28(cat# 102116), anti-CD3 BV605, anti-CD4-PE, anti-CD8- Pe/Cy7, anti-Eomes-Pe/Cy7, anti-CD44-Pacific Blue, anti-CD122-APC, anti-CD62L- APC/Cy7, anti-T-bet-BV421, anti-CTLA4-PE, anti-PD1- BV785, anti-H-2K^b^-Pacific Blue, anti-TNF-α-FITC, anti-IFNγ-APC, anti- EdU-AF647, anti-CD45.1-FITC, anti-CD122-APC, anti-TCF-1-PE anti-CD45.2 APC. We performed multiplex ELISAs using the Biolegend LEGENDplex Assay Mouse Th Cytokine Panel kit (741043). D-Luciferin was purchased from Gold Bio (St. Louis, MO, USA). Flow cytometry was performed on a BD LSR Fortessa (BD Biosciences Franklin Lakes, NJ, USA). Data were analyzed with FlowJo software (Tree Star, Ashland, OR, USA). 

For cell sorting, T cells were purified with anti-CD90.2, or anti-CD4 and anti-CD8 magnetic beads using MACS columns (Miltenyi Biotec, Auburn, CA, USA) prior to cell surface staining. FACS sorting was performed with a BD FACS Aria III cell sorter (BD Biosciences). Cells were sorted into sorting media (50% FBS in RPMI) for maximum viability, or Trizol for RNAseq experiment. FACS-sorted populations were typically of > 95% purity. All cell culture reagents and chemicals were purchased from Invitrogen (Grand Island, NY, USA) and Sigma-Aldrich (St. Louis, MO, USA), unless otherwise specified. For signaling analysis, antibodies against PLCγ1, ERK, IRF-4, Granzyme, Perforin, and β-Actin (total) were purchased from Cell Signaling Technology (Danvers, MA, USA). The primary mouse B-ALL blasts cells [15] were transduced with luciferase, and cultured as described previously [49]. 

### 4.3. Allo-HSCT and GVL Studies 

Lethally irradiated BALB/c mice (800 cGy, split into 2 doses of 400 cGy with 12 h interval between) were injected intravenously with 1 × 10^7^ T cell-depleted bone marrow (_TCD_BM) cells with or without 1 × 10^6^ MACS purified CD3^+^ T cells. Donor T cells were taken from WT (C57Bl/6), WT Ly5.1 (B6.SJL-Ptprc^a^ Pepc^b^/BoyCrl), or *Cat-Tg* mice. For GVL experiments, *B*-cell acute lymphoblastic leukemia (B-ALL) primary blasts [14,15] transduced with luciferase were cultured as described previously, and 1 × 10^5^ luciferase-expressing B-ALL blasts cells were used. Mice were evaluated once a week from the time of leukemia cell injection for more than 60 days post-transplant by bioluminescence imaging using the IVIS 200 Imaging System (Xenogen) [49]. Clinical presentation of the mice was assessed 3 times per week by a scoring system that sums changes in 6 clinical parameters (for each parameter score was ranged from 0–2): posture, activity, fur texture, diarrhea, weight loss, and skin integrity [18]. Mice were euthanized if they lost ≥ 30% of their initial body weight or became moribund. 

### 4.4. Cytokine Production, Cytotoxicity, and EdU Incorporation Assays

On Day 7 post-transplantation, serum from cardiac blood and single-cell suspensions of splenocytes were obtained from allo-transplanted recipients. Serum IFN-γ, TNF-α, IL-5, IL-12, IL-6, IL-10, IL-9, IL-17A, IL-17F, IL-22, and IL-13 levels were determined by multiplex cytokine assays (Biolegend LEGENDplex) [15,17]. Splenocytes taken from allo-transplanted recipients were stimulated with anti-CD3/anti-CD28 (2.5 ug/mL) for 6 h in the presence of Golgiplug (BD Cytofix/Cytoperm Plus kit cat#555028) (1:1000). After incubation, the cells were fixed then permeabilized and stained intracellularly for cytokines (IFN-γ and TNF-α). 

For the proliferation assays, recipient BALB/c mice were transplanted as described above (1 × 10^6^ CD4^+^ and 1 × 10^6^ CD8^+^T cells were mixed 1:1 ratio and 1× 10^7^ WT _TCD_BM), and recipient mice were injected at day 5–6 with 25 mg/kg EdU (20518 from Cayman Chemicals (Ann Arbor, Michigan USA) in PBS. On day 7, the recipient mice were euthanized and lymphocytes from the spleen were obtained. Cells were processed and stained using an EdU click chemistry kit (C10424 from Invitrogen), and also stained for H2K^b^, CD3, CD4, and CD8 to identify donor cells as perversely described [50].

For cytotoxicity assays, luciferase-expressing B-ALL cells were seeded in 96-well flat-bottom plates at a concentration of 3 × 10^5^ cells/ml. D-firefly luciferin potassium salt (75 μg/mL; Caliper Hopkinton, MA, USA) was added to each well and bioluminescence was measured with the IVIS-50 Imaging System. Subsequently, effector cells (MACS-purified) were added at 40:1, 20:1, and 10:1 effector-to-target (E:T) ratios and incubated at 37 °C for 4 h. Bioluminescence in relative luciferase units (RLU) was then measured for 1 min. Cells treated with RIPA lysis buffer were used as a measure of maximal killing. Target cells incubated without effector cells were used to measure spontaneous death. Triplicate wells were averaged and percent lysis was calculated from the data using the following equation: % specific lysis = 100X (spontaneous death RLU–test RLU)/(spontaneous death RLU– maximal killing RLU) [51].

### 4.5. Migration Assays 

Lethally irradiated BALB/c mice were injected intravenously with 1 × 10^7^ WT T cell-depleted bone marrow (_TCD_BM), and a 1:1 mixture of WT (B6-Ly5.1) CD45.1^+^ MACS-purified CD8^+^ and CD4^+^ T cells (checked for 1:1 ratio with flow) with either WT (C57BL/6) CD45.2^+^ cells or *Cat-Tg* CD45.2^+^ cells (total 1 × 10^6^ cells). The donor cells were checked pre-transplant for a 1:1 ratio of donor types, and a 1:1 ratio of CD4:CD8 T cells within each donor type. Seven days post-transplantation, the mice were sacrificed and lymphocytes from the liver, small intestine, and spleen were isolated. Livers were perfused with PBS, dissociated, and filtered with a 70 μm filter. The small intestines were washed in media, shaken in strip buffer at 37 °C for 30 min to remove the epithelial cells, and then washed, before digesting with collagenase D (100 mg/mL) and DNase (1 mg/mL) for 30 min in 37 °C, and followed by filtering with a 70 μm filter. Lymphocytes from the liver and intestines were further enriched using a 40% Percoll gradient. The cells were analyzed for H2K^b^, CD45.1^+^ and CD45.2^+^, CD3^+^, CD8^+^ and CD4^+^ by flow cytometry as described before [15,17]. 

### 4.6. RNA Sequencing 

3 Recipient BALB/c mice for each group were short-term transplanted as described above 1 × 10^6^ CD4^+^ and 1 × 10^6^ CD8^+^T cells were mixed 1:1 ratio and 1× 10^6^ WT _TCD_BM), and at day 7, recipient mice were euthanized and splenocytes were obtained for post-transplant samples. Further, 3 WT or *Cat-Tg* mice also were euthanized and fresh splenocytes were isolated for pre-transplanted samples. CD4^+^ and CD8^+^ T cells from each pre- and post-transplanted mouse were FACS sorted as described above. These cells were all sorted into Trizol and brought to the Molecular Analysis Core (SUNY Upstate) for RNA extraction and library prep, followed by RNA sequencing analysis at the University at Buffalo Genomics Core. We generated RNA sequencing data from four groups for each cell subset (CD4/CD8): WT-pre tx and *Cat-Tg* pre-transplant cells (prior to transplantation); and WT-Day7 post-transplant, *Cat-Tg* Day-7 post-transplant (7 days post-transplantation). We were unable to sort enough donor T cells from the small intestines and liver of the recipient mice that received *Cat-Tg* CD3^+^ T cells. All data were processed and analyzed using the R programming language (Version 4.0.4), the RStudio interface (Version 1.4.1106), and Bioconductor. Transcript abundance of samples was computed by pseudoalignment with Kallisto (version 0.46.2) [52]. Transcript per million (TPM) values were then normalized for each CD4 and CD8 sample separately and fitted to a linear model by empirical Bayes method with the Voom and Limma R packages [53,54] and differential gene expression was defined as a ≥ 2 fold, FDR ≤ 5%; after controlling for multiple testing using the Benjamini-Hochberg method [55] were used for hierarchical clustering and heatmap generation in R. For GO enrichment analysis, the g: Profiler [56] toolset, g:GOSt tool was utilized. Gene Set Enrichment Analysis (GSEA) was performed using cluster Profiler [57] and the Molecular Signatures Database (MSigDB) [23] using the Hallmark pathways collection. Data will be deposited (https://www.ncbi.nlm.nih.gov/geo/) Accession number (GSE180808)

The RNAseq experiment described here was performed as part of the experiment described in other recent publications from our lab [15,58]. Therefore, the data generated for WT pre- and post-transplanted samples (CD4 and CD8) are the same as that shown in the papers mentioned, but here, these data are compared to data for *Cat-Tg* mice. 

### 4.7. Western Blotting 

Cells were lysed in freshly prepared lysis buffer (RIPA buffer (Fisher Scientific cat#PI89900, Waltham, MA, USA) + complete protease inhibitor cocktail (Sigma Aldrich cat#11697498001, St. Louis, MO, USA) and centrifuged at 14,000 rpm for 10 min at 4 °C. Aliquots containing 1 × 10^6^ cells were separated on 12–18% denaturing polyacrylamide gel and transferred to nitrocellulose membranes for immunoblot analysis using specific Abs. 

### 4.8. Histopathological Evaluation

Lethally irradiated recipient mice were transplanted with 1 × 10^7^ T cell-depleted bone marrow cells, and 1 × 10^6^ CD4^+^ and 1 × 10^6^ CD8^+^T cells were mixed 1:1 ratio from WT or *Cat-Tg* mice. On day 7 post-transplantation, recipient mouse livers and smalls intestine were obtained and fixed in 10% neutral buffered formalin, then sectioned and stained with H&E by the Histology Core at Cornell University. Obtained tissues were graded for GVHD by a pathologist (A.M), who was blinded to the study group and disease status. Links for grading criteria: http://surgpathcriteria.stanford.edu/transplant/skinacutegvhd/printable.html, http://surgpathcriteria.stanford.edu/transplant/giacutegvhd/printable.html, http://surgpathcriteria.stanford.edu/transplant/livergvhd/printable.html. Statistical analysis was performed using the Mann–Whitney U test. 

### 4.9. Statistics

All numerical data are reported as means with standard deviation unless otherwise noted in figure legends. Data were analyzed for significance with GraphPad Prism v7. Differences were determined using one-way or two-way ANOVA and Tukey’s multiple comparisons tests, chi-square test or with a student’s *t*-test when necessary. We used Mann–Whitney U test for the analysis of GVHD grades. *p*-values less than or equal to 0.05 are considered significant. All transplant experiments are performed with *n* = 5 mice per group, and repeated at least twice, according to power analyses unless otherwise specified. Mice are sex-matched, and age-matched as closely as possible.

## 5. Conclusions

Currently, this study’s limitation is the use of a mouse model. We are working with structural and medicinal chemists to make specific activators for Wnt/β-catenin pathways. The currently available reagents are Wnt3 ligands [59] or the GSK3β-inhibitors [60]. The primary problems with these activators are that either T cells become over-activated or there is non-specific activation of several other signaling proteins. Therefore, we are currently working to develop our own specific activators.

## Figures and Tables

**Figure 1 cancers-13-03798-f001:**
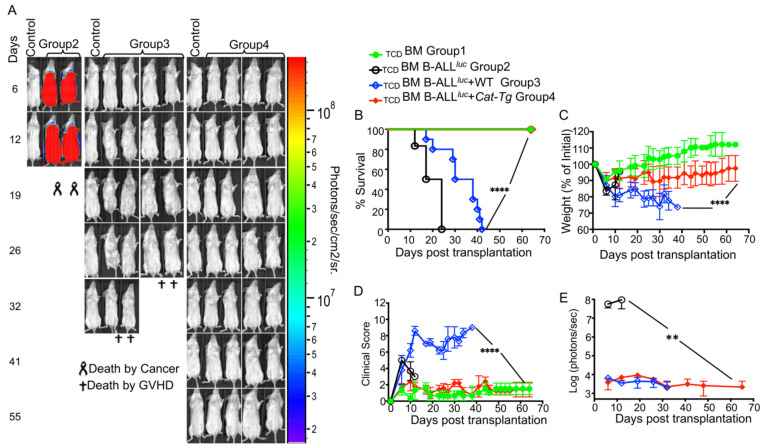
Wnt/β-catenin overexpression in T cells results in retention of GVL effect but avoids GVHD during allo-HSCT. 1 × 10^6^ purified WT or *Cat-Tg* CD3+ T cells were transplanted into lethally irradiated BALB/c recipient mice, along with 1 × 10^7^ T cell-depleted bone marrow cells from WT mice and 1 × 10^5^ B-ALL-*luc* cells. Host BALB/c mice were imaged using the IVIS50 system 3 times a week. Group 1 received T cell-depleted bone marrow only (labeled as _TCD_BM). Group 2 received 1× 10^7^
_TCD_BM from WT mice and 1 × 10^5^ B-ALL-*luc* cells (_TCD_BM+B-ALL *^luc^*^+^). Group 3 was transplanted with 1 × 10^7^
_TCD_BM from WT mice and 1 × 10^6^ purified WT CD3^+^ T cells, along with 1 × 10^5^ B-ALL*^luc^*^+^ cells (_TCD_BM+B-ALL*^luc^*^+^ WT). Group 4 received 1 × 10^7^
_TCD_BM from WT mice and 1 × 10^6^ purified *Cat-Tg* CD3^+^ T cells, along with 1 × 10^5^ B-ALL-*luc*+ cells (_TCD_BM+B-ALL*^luc^*+ *Cat-Tg*). (**A**) Recipient BALB/c mice were imaged using IVIS50 3 times a week. The mice were also monitored for (**B**) survival, (**C**) changes in body weight, and (**D**) clinical score for 65 days post BMT. (**E**) Quantitated luciferase bioluminescence of tumor growth. Statistical analysis for survival and clinical score was performed using log-rank test and one-way ANOVA, respectively. For weight changes and clinical score, one representative of two independent experiments is shown (*n* = 3 mice/group for BM alone; *n* = 5 experimental mice/group for all three other groups) Survival is a combination of two experiments. Note: Control mouse is a naïve mouse used as a negative control for BLI. ** *p* ≤ 0.01; **** *p* ≤ 0.0001.

**Figure 2 cancers-13-03798-f002:**
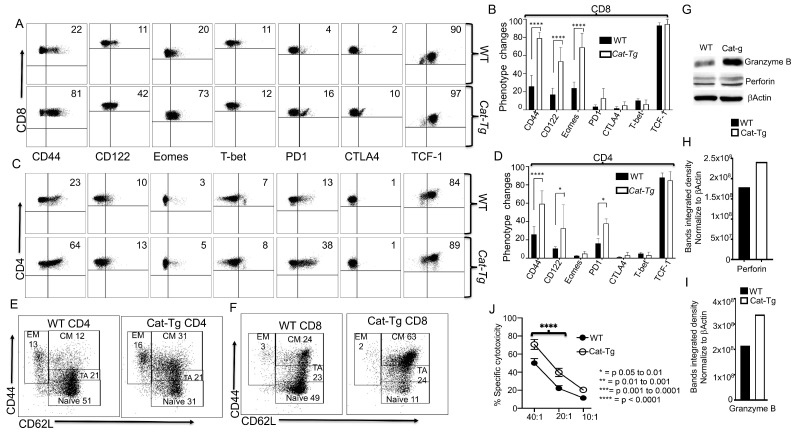
T cells from *Cat-Tg* mice exhibit enhanced T cell IMP phenotypes and enhanced GVL cytotoxicity. Purified CD3^+^ T cells from WT and *Cat-Tg* T cells were examined for expression of CD44, CD122, Eomes, T-bet, PD-1, CTLA-4 and TCF-1 by flow cytometry. These markers were examined for (**A**) CD8^+^ T cells, quantified in (**B**). These markers were also examined for (**C**) CD4^+^ T cells, quantified in (**D**). For (**C**,**D**), combined data from three separate experiments are shown. (**E**) CD4^+^ and (**F**) CD8^+^ T cells from WT and *Cat-Tg* mice were examined for effector memory (EM), central memory (CM), transitioning to activation (TA), and naïve population frequencies. (**G**) Purified T cells were examined for expression of perforin, granzyme B, and β-actin by Western blot. (**H**,**I**) Quantitative analysis of perforin (**H**) and granzyme B (**I**) expression from Western blot data, normalized against β-Actin. (**J**) Ex vivo purified T cells were used in a cytotoxicity assay against primary tumor target B-ALL*luc*+ cells at a 40:1, 20:1, or 10:1 effector to target ratio. Statistical analysis was performed using two-way ANOVA, one-way ANOVA confirmed by Student’s *t*-test, *p*-values are presented. Symbol meanings for *p*-values are: ns—*p* > 0.05; * *p* ≤ 0.05; **** *p* ≤ 0.0001 (*n* = 3 mice per group).

**Figure 3 cancers-13-03798-f003:**
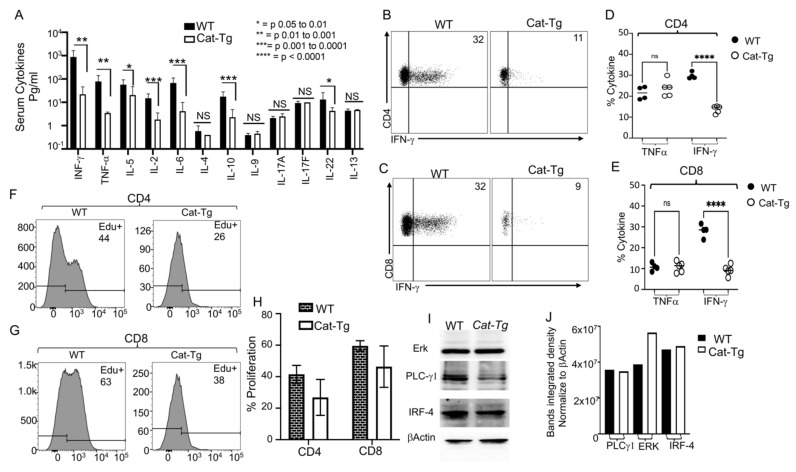
Overexpression of β-Catenin reduces T cell inflammatory cytokine production and proliferation without affecting signaling molecules. (**A**) 1 × 10^6^ purified WT or *Cat-Tg* CD3^+^ T cells were transplanted with _TCD_BM into irradiated BALB/c mice. On day 7 post-allo-HSCT, recipient BALB/c were euthanized and serum cytokines (IFN-γ, TNF-α, IL-2, IL-5, IL-6, IL-4, IL-10, IL-9, IL17A, IL-17F, IL-22, and IL-13) were determined by multiplex ELISA. (**B**,**C**) Intracellular IFN-γ and TNF-α expression by donor CD4 (**B**) and CD8 (**C**) T cells after 6 h stimulation with anti-CD3/anti-CD28 and GolgiPlug, as determined by flow cytometry. (**D**,**E**) Quantified IFN-γ and TNF-α expression for (**B**,**C**). (**F**–**H**) Ex vivo proliferation of donor CD4^+^ or CD8^+^ T cells from *Cat-Tg or* WT mice. Lethally irradiated recipient BALB/c mice were transplanted as mentioned above, with either WT or *Cat-Tg* donor CD3^+^ T cells. Recipient mice were given EdU in PBS i.p. (25 mg/kg in 100 μL) on days 5 and 6 post-transplant. At 7 days post-allotransplantation, recipient mice were sacrificed and examined for proliferation by EdU incorporation via flow cytometry. (**I**) Purified CD3^+^ WT and *Cat-Tg* T cells were examined for total protein expression of ERK, PLCγ-1, and IRF-4 by Western blot. (**J**) Quantitative analysis from Western blots in (I), using ImageLab software to normalize to β-Actin. Symbol meanings for *p* values are: ns—*p* > 0.05; * *p* ≤ 0.05; ** *p* ≤ 0.01; *** *p* ≤ 0.001; **** *p* ≤ 0.0001. (For **A**–**G**, *n* = 5 mice per group, for **H**,**I**, *n* = 3 mice used per group).

**Figure 4 cancers-13-03798-f004:**
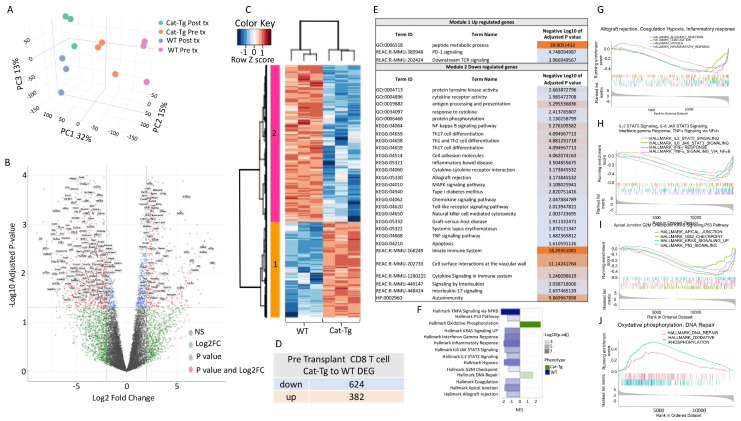
β-catenin overexpression differentially regulates gene expression in CD8^+^ T cells. (**A**) 3D graph of PCA analysis showing clustering of pre-tx and post-tx day 7 CD8^+^ T cells for strain and timepoint by color. All replicates are shown (*n* = 3). (**B**) Volcano plot displaying 1006 differentially expressed genes (FDR ≤ 0.05, log FC ≥ 2) between *Cat-Tg* and WT pre-tx CD8^+^ T cell samples. Positive log fold change corresponds to increased expression in Cat-Tg samples. (**C**) Hierarchical clustering and heat map illustrating expression of genes compared between different groups selected by strain and timepoint. All replicates are shown (*n* = 3) for each group. (**D**) Table showing the number of up- or downregulated DEGs between groups. (**E**) Table showing GO enrichment analysis of DEG of pre-tx CD8^+^ T cells. Using the online tool gProfiler and the ordered g:GOSt query, we assessed which biological processes (BP) were linked to the genes in the different modules from pre-tx CD8^+^ T cells. On the table, adjusted *p*-values were color-coded as light blue for insignificant findings to orange with highest significance. (*n* = 3 mice per group). (**F**) A bar plot showing up- or downregulated pathways based on enrichment scores in GSEA in pre-tx WT or *Cat-Tg* CD8^+^ T cell samples. Negative Normalized Enrichment Score (NES) is an indicator of downregulation and positive NES is an indicator of upregulation of the genes in the corresponding pathway. Color specifies the group (*WT* or Cat-Tg) in which expression is altered (phenotype), color transparency indicates the negative Log10 of adjusted P-value. GSEA enrichment plots of gene clusters that are enriched in *Cat-Tg* cells. (**G**–**J**) GSEA enrichment plots of gene clusters that are enriched in WT. (**G**) Hallmark of Allograft rejection, Hallmark of Coagulation, Hallmark of Hypoxia, Hallmark of Inflammatory response, (**H**) Hallmark of IL-2 STAT5 Signaling, Hallmark of IL-6 JAK STAT3 Signaling, Hallmark of Interferon-gamma Response, Hallmark of TNFA Signaling via NF-kb, (**I**) Hallmark of Apical Junction, Hallmark of G2M Checkpoint, Hallmark of KRAS Signaling, Hallmark of P53 Pathway, and (**J**) Hallmark of DNA Repair and Hallmark of Oxidative Phosphorylation.

**Figure 5 cancers-13-03798-f005:**
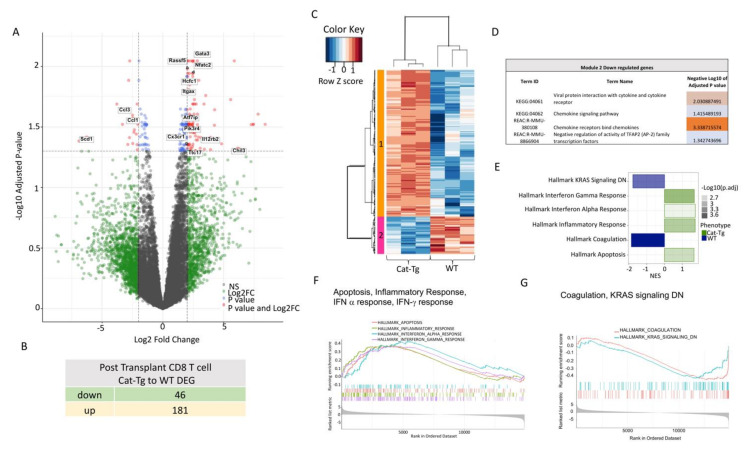
β-catenin overexpression differentially regulates gene expression in post-transplanted donor CD8^+^ T cells. (**A**) Volcano plot displaying 227 differentially expressed genes (FDR ≤ 0.05, logFC ≥ 2) between *Cat-Tg* and WT post-tx CD8^+^ T cell samples. Positive log fold change corresponds to increased expression in *Cat-Tg* samples. (**B**) Table showing the number of up- or downregulated DEGs between groups. (**C**) Hierarchical clustering and heat map illustrating expression of genes compared between different groups selected by strain and timepoint. All replicates are shown (*n* = 3) for each group. (**D**) Table showing GO enrichment analysis of DEGs of post-tx CD8^+^ T cells. Using the online tool gProfiler and the ordered g:GOSt query, we assessed which biological processes (BP) were linked to the genes in the different modules from post-tx CD8^+^ T cells. On the table, adjusted *p*-values were color-coded as light blue for insignificant findings to orange with highest significance. (*n* = 3 mice per group). (**E**) A bar plot showing up- or downregulated pathways based on enrichment scores in GSEA in post-tx WT or *Cat-Tg* CD8^+^ T cell samples. Negative Normalized Enrichment Score (NES) is indicator of downregulation and positive NES is indicator of upregulation of the genes in the corresponding pathway. Color specifies the group (*WT* or Cat-Tg) in which expression is altered (phenotype), color transparency indicates the negative Log10 of adjusted *p*-value. (**F**,**G**) GSEA enrichment plots of gene clusters that are enriched in *Cat-Tg*, including Hallmark of Apoptosis, Hallmark of Inflammatory Response, Hallmark of Interferon-alpha response, and Hallmark of Interferon gamma response. (**G**) GSEA enrichment plots gene clusters that are enriched in WT, including Hallmark of Coagulation and Hallmark of KRAS signaling DN.

**Figure 6 cancers-13-03798-f006:**
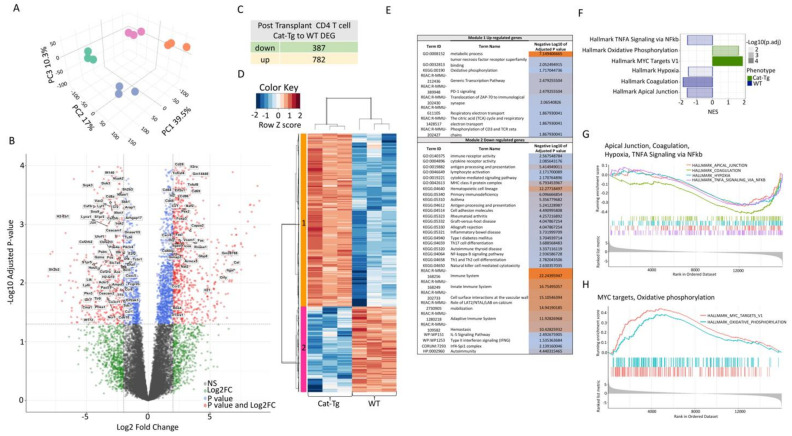
β-catenin overexpression differentially regulates gene expression in CD4^+^ T cells. (**A**) 3D graph of PCA analysis showing clustering of pre-tx and post-tx day 7 CD4^+^ T cells for strain and timepoint by color. All replicates are shown (*n* = 3). (**B**) Volcano plot displaying 1169 differentially expressed genes (FDR ≤ 0.05, logFC ≥ 2) between *Cat-Tg* and WT pre-tx CD8^+^ T cell samples. Positive log fold change corresponds to increased expression in Cat-Tg samples. (**C**) Table showing the number of up- or downregulated DEGs between groups. (**D**) Hierarchical clustering and heat map illustrating expression of genes compared between different groups selected by strain and timepoint. All replicates are shown (*n* = 3) for each group. (**E**) Table showing GO enrichment analysis of DEG of pre-tx CD4^+^ T cells. Using the online tool gProfiler and the ordered g:GOSt query, we assessed which biological processes (BP) were linked to the genes in the different modules from pre-tx CD4^+^ T cells. On the table, adjusted *p*-values were color-coded as light blue for insignificant findings to orange with highest significance. (*n* = 3 mice per group). (**F**) A bar plot showing up- or downregulated pathways based on enrichment scores in GSEA in pre-tx WT or *Cat-Tg* CD4^+^ T cell samples. Negative Normalized Enrichment Score (NES) is indicator of downregulation and positive NES is indicator of upregulation of the genes in the corresponding pathway. Color specifies the group (*WT* or Cat-Tg) in which expression is altered (phenotype), color transparency indicates the negative Log10 of adjusted *p* value. (**G**) GSEA enrichment plots of gene clusters that are enriched in WT, including Hallmark of Apical Junction, Hallmark of Coagulation, Hallmark of Hypoxia, and Hallmark of TNF-a Signaling via NF-kb. (**H**) GSEA enrichment plots of gene clusters that are enriched in *Cat-Tg*, including Hallmark of MYC targets and Hallmark of Oxidative phosphorylation.

**Figure 7 cancers-13-03798-f007:**
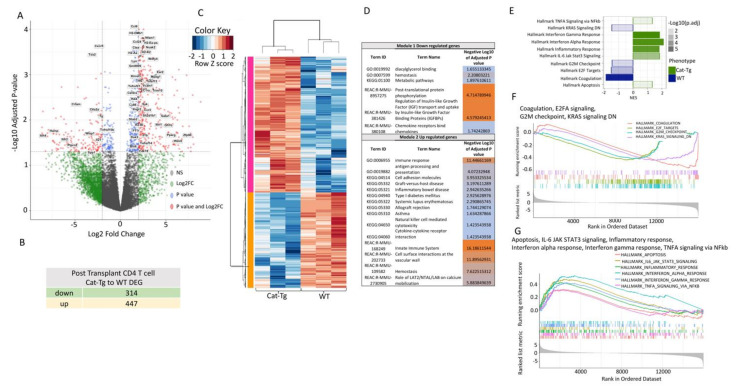
β-catenin overexpression differentially regulates gene expression in post-transplanted donor CD4^+^ T cells. (**A**) Volcano plot displaying 761 differentially expressed genes (FDR ≤ 0.05, logFC ≥ 2) between *Cat-Tg* and WT post-tx CD4^+^ T cell samples. Positive log fold change corresponds to increased expression in Cat-Tg samples. (**B**) Table showing the number of up- or downregulated DEGs between groups. (**C**) Hierarchical clustering and heat map illustrating expression of genes compared between different groups selected by strain and timepoint. All replicates are shown (*n* = 3) for each group. (**D**) Table showing GO enrichment analysis of 761 DEG of post-tx CD4^+^ T cells. Using the online tool gProfiler and the ordered g:GOSt query, we assessed which biological processes (BP) were linked to the genes in the different modules from post-tx CD4^+^ T cells. On the table, adjusted *p*-values were color-coded as light blue for insignificant findings to orange with highest significance. (*n* = 3 mice per group). (**E**) A bar plot showing up- or downregulated pathways based on enrichment scores in GSEA in post-tx WT or *Cat-Tg* CD4^+^ T cell samples. Negative Normalized Enrichment Score (NES) is indicator of downregulation and positive NES is indicator of upregulation of the genes in the corresponding pathway. Color specifies the group (*WT* or Cat-Tg) in which expression is altered (phenotype), color transparency indicates the negative Log10 of adjusted *p* value. (**F**) GSEA enrichment plots of gene clusters that are enriched in WT, including Hallmark of Coagulation, Hallmark of E2FA signaling, Hallmark of G2M checkpoint, Hallmark of KRAS signaling DN. (**G**) GSEA enrichment plots of gene clusters that are enriched in *Cat-Tg,* including Hallmark of Apoptosis, Hallmark of IL-6 JAK STAT3 signaling, Hallmark of Inflammatory response, Hallmark of Interferon-alpha response, Hallmark of Interferon-gamma response, and Hallmark of TNFA signaling via NFκb.

**Figure 8 cancers-13-03798-f008:**
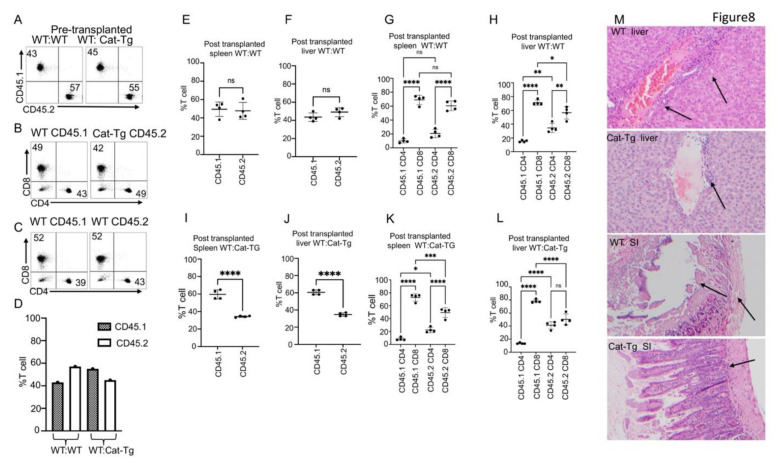
Wnt/β-catenin overexpression specifically affects CD8^+^ T cell functions. (**A**) Pre-allo-transplanted MACS-purified WT(CD45.1) and *Cat-Tg* (CD45.2) cells, and WT(CD45.1) and WT(CD45.2) (as a control) were mixed at a 1:1 ratio. (**B**) Cell mixture of *Cat-Tg* (CD45.2) cells, and WT(CD45.1) was stained for CD4 and CD8 surface markers. (**C**) Cell mixture of WT (CD45.2) cells, and WT(CD45.1) was stained for CD4 and CD8 surface markers. (**D**) Quantification data of pre-transplanted CD4 and CD8 T cells from CD45.1 WT and CD45.2 WT mice were examined. We also examined CD4 and CD8 T cells from CD45.1 WT and CD45.2 *Cat-Tg* mice. (**E**) Quantified data from 7 days post-transplantation, when recipient mice were euthanized and examined for donor T cells (CD45.1 or CD45.2) in spleen from the WT(CD45.1): WT(CD45.2) mice. (**F**) Quantified data for donor cells in recipient liver from the WT(CD45.1): WT (CD45.2) mice at day 7 post-BMT, looking at CD45.1/CD45.2. (**G**) Donor CD4 and CD8 T cells in recipient spleen from the WT(CD45.1): WT(CD45.2) mice at day 7 post-BMT were compared. (**H**) Donor CD4 and CD8 T cells in recipient liver from the WT(CD45.1): WT(CD45.2) mice at day 7 post-BMT were compared. (**I**) Quantification of donor cells from the WT(CD45.1): *Cat-Tg* (CD45.2) mice at day 7 post-BMT in recipient spleen were compared, looking at CD45.1/CD45.2. (**J**) Quantification of donor T cells in recipient liver from the WT(CD45.1): Cat-Tg (CD45.2) mice at day 7 post-BMT, looking at CD45.1/CD45.2. (**K**) Comparison of donor CD4 and CD8 T cells in recipient spleen from the WT(CD45.1): *Cat-Tg* (CD45.2) mice at day 7 post-BMT. (**L**) Comparison of donor CD4 and CD8 T cells in recipient liver from the WT(CD45.1): *Cat-Tg* (CD45.2) mice at day 7 post-BMT. (**M**) On day 7 post-allo-HSCT, small intestines and liver were examined by (H&E) staining for tissue damage. Arrows show lymphocyte infiltration and tissue damage. Statistical analysis was performed using one-way ANOVA. Mann–Whitney U test, *p*-value presented with the Figure Symbol meaning for *p* values are: ns—*p* > 0.05; * *p* ≤ 0.05; ** *p* ≤ 0.01; *** *p* ≤ 0.001; **** *p* ≤ 0.0001. (For **A**–**E**, *n* = 5 mice per group, repeated twice, one representative is shown. For **F**,**G**, *n* = 5 mice used per group). scale bar or magnification 20×:

## Data Availability

For all other reagents please contact Mobin Karimi, Department of Microbiology and Immunology. MTA rules will be applied. SUNY Upstate Medical University, 766 Irving Ave Weiskotten Hall Suite 2281, Syracuse, NY 13210 (karimim@upstate.edu); RNA sequences data are deposited in https://www.ncbi.nlm.nih.gov/geo; Accession number: For all other reagents please contact the lead contacts; All resources are available to anyone by request to lead contact karimim@upstate.edu; No software products, custom code, or algorithms were developed for this manuscript.

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
