# Peer review of "Human Wnt/β-Catenin Regulates Alloimmune Signaling during Allogeneic Transplantation"

_cancers, 2021, doi:10.3390/cancers13153798_

Round 1

Reviewer 1 Report

Mammadli's text is interesting because it appropriately describes in a pre-clinical setting the role of Wnt / B catenin in the regulation of the immunological response after mismatched allogeneic transplantation. The abstract and introduction sections are concise, the methods part must be better detailed, the results section is well explained and the discussion congruent with the results obtained. references are up to date

Minor issues:

1) authors should report how murine lymphocytes were transduced,

2) The acronym T bet is not explicit

3) The authors should also indicate the future direction of this study, in other words how it is possible to pharmacologically increase the expression of Wnt / B catenin in humans after allogeneic transplantation. 

Author Response

Reviewer #1

Comments to the Author

Mammadli's text is interesting because it appropriately describes in a pre-clinical setting the role of Wnt / B catenin in the regulation of the immunological response after mismatched allogeneic transplantation. The abstract and introduction sections are concise, the methods part must be better detailed, the results section is well explained and the discussion congruent with the results obtained. references are up to date

Minor issues:

Reviewer #1

  • authors should report how murine lymphocytes were transduced,

Response to Reviewer #1: We thank the reviewer for the suggestion, in the revised manuscript we have added a section on how these mice were generated, and highlighted Page #22 line (439-446)

  • The acronym T bet is not explicit

Response to Reviewer #1: We thank the reviewer for critically reviewing our manuscript. We have updated the acronym for T-bet at page #7 line (144-145)

  • The authors should also indicate the future direction of this study, in other words how it is possible to pharmacologically increase the expression of Wnt / B catenin in humans after allogeneic transplantation.

Response to Reviewer #1 We thank the reviewer for keeping our research interesting. In the revised manuscript, on page 22 line (437-422) we added a section on limitations and future directions. Please stay tuned for our upcoming manuscript. Also, on page 22 we added a section on resources availability, line (444-452)

Reviewer 2 Report

The Mammadli and Harris text is very interesting for the scientific community dealing with stem cell transplantation and its immunological complications. The authors demonstrate that the overexpression of Wnt / B-catenin in lymphocytes of HLA-mismatch mice generates the maintenance of the GvL effect while the GvHD reaction appears to be greatly mitigated. In particular, the authors demonstrate how the overexpression of Wnt / B-catenin changes the gene profile of CD4 + and CD8 + lymphocytes, while the expression of TCR does not appear to be modified. The introduction is clear and oriented, the materials and methods section is concise, the results are detailed and the discussion is focused on the results obtained

The statistical part is not evaluable.

Minor issues

1) it is not shown how C57Bl / 6 mice were transduced with the B-catenin gene

2) T-bet must be specified in the text

3) Should the authors conclude with future perspectives more specifically, in particular which drugs are licensed for humans with this activity, if any? 

Author Response

Reviewer #2 - Comments to the Author

Comments and Suggestions for Authors

The Mammadli and Harris text is very interesting for the scientific community dealing with stem cell transplantation and its immunological complications. The authors demonstrate that the overexpression of Wnt / B-catenin in lymphocytes of HLA-mismatch mice generates the maintenance of the GvL effect while the GvHD reaction appears to be greatly mitigated. In particular, the authors demonstrate how the overexpression of Wnt / B-catenin changes the gene profile of CD4 + and CD8 + lymphocytes, while the expression of TCR does not appear to be modified. The introduction is clear and oriented, the materials and methods section is concise, the results are detailed and the discussion is focused on the results obtained

The statistical part is not evaluable.

Minor issues

  • it is not shown how C57Bl / 6 mice were transduced with the B-catenin gene

Response to Reviewer #2 We thank the reviewer for the suggestion, in the revised manuscript we have added a section, on how these mice were generated and highlighted Page #22 line (439-446)

2) T-bet must be specified in the text

Response to Reviewer #2 We thank the reviewer for critically reviewing our manuscript. We have updated the acronym for T-bet at page #7 line (144-145)

  • Should the authors conclude with future perspectives more specifically, in particular which drugs are licensed for humans with this activity, if any?

Response to Reviewer #2 We thank the reviewer for keeping our research interesting. In the revised manuscript, on page 22 line (437-422) we added a section on limitations and future directions. Please stay tuned for our upcoming manuscript. Also, on page 22 we added a section on resources availability, line (444-452)
